# Reliable Field Assessment of Proliferative Kidney Disease in Wild Brown Trout, *Salmo trutta*, Populations: When Is the Optimal Sampling Period?

**DOI:** 10.3390/pathogens11060681

**Published:** 2022-06-14

**Authors:** Aurélie Rubin, Christyn Bailey, Nicole Strepparava, Thomas Wahli, Helmut Segner, Jean-François Rubin

**Affiliations:** 1Centre for Fish and Wildlife Health, University of Bern, 3012 Bern, Switzerland; christyn.bailey@outlook.com (C.B.); nicole.strepparava@bluewin.ch (N.S.); thomas.wahli@vetsuisse.unibe.ch (T.W.); helmut.segner@vetsuisse.unibe.ch (H.S.); 2La Maison de la Rivière, 1131 Tolochenaz, Switzerland; jf.rubin@maisondelariviere.ch; 3Land, Nature, Environment Institute, University of Applied Sciences and Arts Western Switzerland, 1202 Geneva, Switzerland; 4Fish Immunology and Pathology Laboratory, Animal Health Research Centre (CISA-INIA), 28130 Madrid, Spain

**Keywords:** proliferative kidney disease, *Salmo trutta*, *Tetracapsuloides bryosalmonae*, sampling time point, water temperature, degree days

## Abstract

Proliferative kidney disease (PKD), caused by the myxozoan parasite *Tetracapsuloides bryosalmonae*, is suspected to contribute to the decline of wild brown trout *Salmo trutta* populations. Different factors need to be taken into consideration for PKD outbreaks. Among them, water temperature appears as a main driver of the disease. To understand the epidemiology and impact of the disease on wild fish populations, reliable sampling approaches to detect the presence of *T. bryosalmonae*-infected fish are needed. This study aimed to characterize the seasonal variation of the prevalence of *T. bryosalmonae*-infected fish in brown trout populations in two small streams with differing temperature regimes between upstream and downstream sites. As water temperature is known to influence PKD manifestation in brown trout, we hypothesized that the number of *T. bryosalmonae*-positive fish, as well as their seasonal distribution, will vary between upper and downstream parts of the two streams. Since, in field studies, results can strongly vary across years, we extended the study over a 3-year-period. The number of infected fish and the intensity of infection were assessed by histology. The results confirmed the hypothesis of pronounced temporal- and site-related differences in the percentage of PKD-positive fish and the intensity of the infection. Comparison of water temperatures (total degree days as well as the number of days with a daily mean temperature ≥15 °C) with PKD data indicated that temperature was the driving factor for the temporal development and the intensity of the infection. A mean of 1500 degree days or 30 days with a daily mean temperature ≥15 °C was required before the infection could be detected histologically. From our findings, recommendations are derived for a water temperature-driven sampling strategy campaigns that enables the detection of PKD infection and prevalence in wild brown trout populations.

## 1. Introduction

Brown trout (*Salmo trutta*, L.) populations are declining in Switzerland [1,2,3,4]. A combination of different factors might explain this diminution (e.g., reproductive failure, pollutions, inadequate fish management, increase of water temperature, genetic admixture caused by illegal translocations, etc.) [1,2,3,4,5]. Among them, proliferative kidney disease (PKD) is also involved in this decrease [6,7,8]. The infection is caused by the myxozoan parasite *Tetracapsuloides bryosalmonae* [9]. The parasite possesses a heteroxenus life cycle, with salmonid fish species as the vertebrate host and freshwater bryozoans as the invertebrate host [10,11,12]. In bryozoans, covert and overt infection stages can occur. During a covert infection, the parasite is under its non-virulent single cell form, associated with the body wall of the bryozoan [11]. This stage can be found throughout the year [13]. If the necessary conditions are met (food resources and temperature), overt infection can occur [14]. Multicellular sacs of *T. bryosalmonae* spores develop in bryozoans and are then undulatingly released into the water, with major peaks in spring and summer [11,12,15]. If the parasite come into contact with a susceptible salmonid (e.g., brown trout; marble trout, *Salmo marmoratus*; grayling, *Thymallus thymallus*; or rainbow trout *Oncorhynchus mykiss*), it infects the fish through skin and gills [16,17,18]. In the fish host, the parasite targets the kidney and can induce pronounced renal pathology, and mortalities. The disease severity is strongly influenced by environmental conditions, in particular, temperature [19,20,21,22]. The fact that water temperature affects the development and intensity of PKD in salmonids has been shown for rainbow trout, e.g., [19,20,23,24,25,26,27,28], and brown trout, e.g., [8,27,29,30,31,32]. Laboratory studies using constant temperature regimes provided evidence that water temperatures ≥15 °C are associated with a strong increase in disease severity and disease-related mortalities, e.g., [17,19,20,27,33,34]. Indeed, on this host–parasite system, water temperature could affect the fish, the bryozoans, or the parasite itself. Temperature impacts the kinetic of *T. bryosalmonae* proliferation in the kidney, the lesions severity and their distribution in the renal tissue [19,20], and the strategy of the fish immune response [26], as well as the rate of spore production and transmission [33]. In the invertebrate host, temperature also induces, speeds up, and prolongs the production of infective spores and enhances bryozoan growth [12].

The question is how such laboratory findings can be transferred to the field situation, which is characterized by daily and seasonal temperature fluctuations. Moreover, under field conditions, additional stressors, such as limited food supply or poor water quality, may influence the response of the fish to the parasite [27,31,35].

Given the fact that PKD can impact wild salmonid stocks, the availability of robust sampling approaches to reliably assess the PKD status of fish populations is essential. Successful field diagnosis depends mainly on three parameters: (1) appropriate sample size that avoids or minimizes false negatives, (2) reliable detection methods, and (3) sampling time point. In this study, we focused on the last factor. Indeed, the time point for field sampling campaigns should be well planned. Given the temperature dependence of PKD manifestation, we hypothesize that early or late in the year, when field water temperatures are low, only a few individuals may be infected and/or display clear signs of infection, and, therefore, field sites where PKD is present may be falsely classified to be PKD-negative. On the other hand, when water temperatures are high, i.e., during the summer season, infected fish may have already been eliminated from the population due to PKD-induced mortality, which again would result in an underestimation of the true PKD presence in the rivers. Therefore, the present study aimed to follow the manifestation of PKD in wild brown trout populations concerning the seasonal variations in water temperature. The expected outcome of this study is to identify the suitable period for reliable PKD diagnosis which is based on the actual water temperature regime at a field site. To this end, we investigated two streams that host PKD-positive brown trout populations (Rubin and Wahli, unpublished data), and which show different temperature regimes along the river course. The water temperatures were continuously recorded utilizing a data logger. Young-of-the-year (YOY) brown trout were selected for sampling, as this life stage appears to be most susceptible to the parasite [16,29,36,37]. A sampling period of 3 consecutive years was selected as this is necessary to account for the inter-annual variability [38]. For PKD diagnosis, histology was employed, as the technique is suited for practical field sampling, and it enables to estimation of both disease prevalence and infection intensity.

## 2. Results

The results of the histology-based detection of *T. bryosalmonae*-infected fish (expressed as the percentage of *T. bryosalmonae*-infected fish and the infection intensity per site and time point) and water temperature values (expressed as the degree days (dd) and the number of days with a daily mean temperature ≥15 °C (ndays15)) are summarized in Table 1.

In the first study year, no PKD-positive fish were found in the upstream sites of the Boiron (BM1) and the Venoge (V1). This was true for all five sampling dates from June to November. For the downstream sites of the Boiron (BM2) and the Venoge (V2), infected fish were not observed in June and July. PKD-positive fish were detected for the first time in August, with 68% positive fish at BM2 and 40% at V2. The infection intensity had a mean value of 3.9 at BM2 and of 2.9 at V2. In September, the values were comparable to August, with 68% *T. bryosalmonae*-infected fish and an infection intensity of 3.2 at BM2. At the V2 site, the next sampling after August was conducted in October, yielding 44% positive fish and an infection intensity of 1.8. In November, 31% of sampled fish were assessed as *T. bryosalmonae*-positive for BM2 and 44% for V2. While the percentage of positive fish did not decline from August to November, a decrease was evident in the infection intensity, suggesting a declining number of parasites per fish.

The results of the second study year were comparable to those of year 1. Again, no *T. bryosalmonae*-infected fishes were detected at the upstream sites, BM1 and V1. In the downstream sites, positive fish were not detected in July, but from August onwards. The percentage of infected fish reached 88% in September for BM2 and 38% for V2. The corresponding intensity values were 4.6 in BM2 and 2.8 in V2.

In the third sampling year, at BM2, a high percentage of positive fishes (68%), with a high infection intensity (3.3), were already detected in July, whereas V2 was still PKD-negative in July. In August, the populations at both downstream sites contained infected fishes. Moreover, in August, for the first time, a *T. bryosalmonae*-positive fish was detected at the upstream BM1-site. However, since this was the only isolated observation of a PKD-positive fish at this site during the whole study period, we assume that this was a migratory fish coming from a downstream location.

The samplings showed a clear seasonal pattern of the PKD-infected fish percentage, which raises the question of the potential driving factors. One such parameter might be the water temperature. To examine the possible relation between water temperature and PKD, we continuously monitored water temperatures at the study sites and expressed the data as (a) degree-days (dd), i.e., the sum of daily temperature values, and (b) the number of days with a mean water temperature ≥15 °C (ndays15). The cut-off value of ≥15 °C was selected since we know from previous studies that PKD-induced pathologies and mortalities are strongly intensified at temperatures above 15 °C e.g., [19,20,34].

During the three study years, the water temperature regime showed a distinct seasonal pattern (Table 1). In the two streams, the dd values were higher at the downstream sites compared to the upstream sites. Differences between sites and streams were also evident for the number of days with mean water temperatures of ≥15 °C. At BM1, water temperature reached mean values of ≥15 °C at 20–49 days, whereas at BM2 this was the case for 57–88 days. For instance, in the first study year, brown trout at BM1 experienced 20 days of mean water temperatures of ≥15 °C (maximum of 9 consecutive days with mean water temperature ≥15 °C), but brown trout at the downstream site BM2 were exposed to such temperatures for 57 days (maximum of 25 consecutive days with mean water temperature ≥15 °C). In the third study year, the summer was very hot and dry, resulting in low water levels and particularly high-water temperatures (71 consecutive days with mean water temperature ≥15 °C obtained at BM2). The Venoge generally had fewer days with elevated water temperature. At the V1 upstream site, no single day with a mean water temperature of ≥15 °C was recorded. At the downstream site V2, the respective values were 63 and 59 ndays15 for years 1 and 2.

The relationship between PKD values (percentage of *T. bryosalmonae*-infected fish and infection intensity) and temperature data (dd and ndays15) for the two streams was assessed using Pearson’s correlation. A positive linear correlation was found between the percentage of *T. bryosalmonae*-infected fish and dd (r = 0.476). For ndays15, the correlation was even stronger, with a Pearson’s coefficient value of 0.857. Furthermore, infection intensity and dd were positively correlated, although the coefficient was relatively low (r = 0.245). On the contrary, the correlation was stronger with ndays15 (r = 0.556). In addition, when combining data from all sites, Student’s *t*-test showed that the dd and ndays15 mean was significantly higher for *T. bryosalmonae* positive fish (respectively 2328 dd and 62.6 ndays15) than for PKD-free trout (respectively 1762 dd and 19.3 ndays15) (Figure 1), suggesting an influence of water temperature on the infection.

When combining the data of all three years in BM2 and V2, sites were assessed as PKD-negative with maximum temperature values of, respectively, 1417 dd or 28 ndays15 at BM2, and 1397 dd or 13 ndays15 at V2. When these values were surpassed, PKD-positive fish could be detected by histological diagnosis (Figure 2).

This suggests that the threshold for histology-based detection of PKD in wild brown trout populations appears to be between 1400 and 1600 dd or 28 to 34 ndays15. Hence, 1500 dd and 30 ndays15 were assessed as a mean.

## 3. Discussion

When planning effective fieldwork campaigns, several questions need to be addressed; in particular, the number of samples [8], choice of diagnostic methods [19,20,37], and selection of sampling periods [28,29,30,31]. In this study, we investigated to what extent the sampling season influences the detectability of the presence of PKD infections in wild brown trout populations by means of histology. The higher the percentage of infected fish is in a population, the higher is the likelihood to detect the infection, particularly when only small sample sizes are possible [8]. Wahli et al. [8] showed that a sample of 20 fish is adequate for the detection of PKD when the prevalence in the examined population is equal or higher than 10%. To accurately measure prevalence below 10%, substantially higher sample sizes would be needed. However, if it comes to small streams carrying relatively small brown trout populations, such high sample sizes cannot be justified from ecological and ethical reasons. Therefore, we decided that the accuracy that is achievable with a sample size of 25 fish is sufficient for the purpose of our study.

Typical methods for PKD diagnosis comprise histology and qPCR [19,20,37]. Differences between the microscopic technique (histology) and the molecular technique (qPCR) were investigated in previous papers [19,20,37]. In these studies, qPCR was shown to detect parasite material for a longer period in the kidneys of PKD infected rainbow trout in contrast to microscopic technique. When using histology as a diagnostic method, the likelihood to detect parasites on a tissue section increases with the intensity of the infection. However, in fish with very low infection levels and few parasites present in the kidney, histological examination may have missed the parasites, particularly, as only one slide was analyzed. In contrast, qPCR methodology relying on whole tissue extraction, is less dependent on parasite quantity and on heterogeneous distribution of the parasites within the target organ. Nevertheless, use of histology provides some advantages, as an easier field procedure for the fixation of sampled fish. Moreover, this method is the only one able to assess the severity of the pathological response and the state of the disease, as presence of fibrous tissue revealing begin of recovery and was thus used here.

Since the water temperature regime is the main driver of PKD manifestation in brown trout, we hypothesized that the number of infected fish in a population, as well as the infection intensity, will co-vary with the seasonal variation of water temperature at a given field study site. The results of our study are in agreement with this hypothesis.

Studies conducted in the laboratory with constant temperatures showed that *T. bryosalmonae* infection in brown trout has a pattern of temporal variation. For example, in the study of Strepparava et al. [33], a sharp increase in parasites’ number was detected after exposure, followed by a plateau phase, and, finally, a slow decrease of parasites was observed. In lab studies, however, initial exposure to parasites is synchronous and further infection is not possible as the parasite is not transmitted from fish to fish [13,18,39]. In the wild the situation is different: diurnal and seasonal water temperature regimes are fluctuating, long time exposure to new parasites could appear, co-stressors might be present, etc. Despite evidence that this seasonal distribution of PKD cases may depend on temperature [8,21,29,30,31,32,40], it has not yet been fully elucidated whether this course of PKD in the field is directly linked to the course of the water temperature over the year.

Our results showed that water temperature is the main factor for PKD infection. Differing temperature regimes appear between upstream and downstream sites in the two rivers, as well as water quality as observed in Rubin et al. [31]. PKD-positive fish were detected in the two downstream sites from the Boiron and the Venoge, as regularly previously observed (Rubin and Wahli, unpublished data). A positive correlation between PKD values (percentage of *T. bryosalmonae*-infected fish and infection intensity) and temperature data (dd and ndays15) was observed. In early summer, no parasites were detected in the evaluated fish, while the highest percentage of *T. bryosalmonae*-infected fish were detected in the period from late August to the beginning of September for years 1 and 2. Thus, samplings taken too early in the season, (e.g., BM2 05 June and 07 August samplings of year 1) could miss the infection. At these dates, either infection could not have been taken place yet or not have developed enough to be detected by histology. However, later in the season, when 1500 dd were surpassed, *T. bryosalmonae*-infected fish were detected. Thus, sites might be wrongly classified as PKD negatives if the sampling period is not adequately chosen. On the contrary, too late in the season, as in the November samples, a large number of animals might have died because of the infection, or parasites might have already been excreted or eliminated by the fish immune system [11,19,20,21,41], resulting in low parasite number. In addition, the correct timepoint for fish sampling is not linked to a month, but rather to the course of water temperature, as it happens in the July BM2 sampling of year 3. Indeed, PKD-positive fish were already observed at this date, as a result of the extraordinary temperature conditions in this particular year (1627 dd). Hence, a very hot summer may lead to a shift in the period for PKD sampling. With rising temperature due to global warming [42], this phenomenon may become more and more frequent.

In the wild, parasite infection and PKD-related mortalities appear throughout the year and seem to be more noticeable in summer and the beginning of autumn [30,31], which corresponds to our observations. Wahli et al. [29] identified PKD-positive fish between June to November, with major peaks in August and September. Dash and Vasemägi [43] detected the highest prevalence in August. Palikova et al. [28] found that the highest parasite loads were observed in September in rainbow trout raised on fish farms, but under natural water temperature conditions. These authors also observed that only sporadically single *T. bryosalmonae* parasites were present in fish kidneys sampled in November and December, indicating a significant decline of the parasite presence during the winter months, which corresponds to our observations. In other studies, PKD-positive fish were observed in streams where the water temperature exceeded 15 °C for 31 days [40], 39 days [22], and, respectively, 48, 83, and 80 days along the three investigation years [44]. Thus, these studies support our results, even if we obtained a smaller ndays15 (mean of 30 days) for the detection of *T. bryosalmonae* positive fish. The Swiss Fischnetz project [45] predicted 14 to 28 days at 15 °C for PKD infection. In a previous YOY sampling campaign encompassing 45 sites, Rubin et al. [31] observed that PKD prevalence ranging from 4 to 100% were always found in sites with a minimum of 1900 dd.

## 4. Materials and Methods

### 4.1. Fish Sampling

Sampling campaigns were carried out from 2013 to 2015 (2013 = year 1; 2014 = year 2; 2015 = year 3). Fieldwork was performed at the Boiron de Morges (BM) and the Venoge (V), two streams of the Canton of Vaud (Switzerland) (Figure 3). Two sites were selected in BM (upstream BM1 46.49812° N 6.43836° E, downstream BM2 46.49567° N 6.47410° E), and two sites in V (upstream V1 46.62722° N 6.42770° E, downstream V2 46.55494° N 6.53233° E). Five electrofishing campaigns were carried out between June and November of year 1. BM sites were also sampled in July and September of year 2 and in July and August of year 3. V sites were tested in September of year 2 as well. Whenever possible, 25 fish were caught. These individuals were selected based on their total length in order to correspond to the average size class for young of the year brown trout from typical Swiss midland streams (<100 mm) (Rubin, unpublished data).

### 4.2. Histological Analysis

After capture, YOY were euthanized with MS222^®^ (3-aminobenzoic acid ethyl ester, 300 mg l-1, Argent Chemical Laboratories, Redmont, WA, USA), fixed, and stored in containers containing 4% buffered formalin. Kidneys were removed from the carcasses in the laboratory, embedded in paraffin, and cut following routine histological methods. Histological slides were stained with haematoxylin and eosin. One section of a full-length kidney per animal was analyzed for the presence of *T. bryosalmonae*, determined through its typical cell structure (spores with four polar capsules). The infection intensity (estimation of the numbers of observed parasites) was assessed for each kidney sample with a microscope (Olympus BX41). For this purpose, a scoring system from 0 (no parasite) to 6 (at least 10 parasites per high power field with 400× magnification) was used (Figure 4) following Bettge et al. [19] and Schmidt-Posthaus et al. [22]. A fish was classified as *T. bryosalmonae*-infected if at least one parasite was detected in the analyzed kidney section. The infection intensity per site was obtained as the addition of infection scores divided by the number of PKD-positive fish. The percentage of *T. bryosalmonae*-infected fish was determined as the number of infected individuals divided by the total number of sampled fish at a particular site and time-point as described in Bush et al. [46].

### 4.3. Water Temperature

Water temperature was recorded at all sites every 15 min with loggers (HOBO^®^ Water Temp Pro v2 Data Logger, Onset, Cape Cod, MA, USA). Temperature data started on the 1st of March, as from this date on fry hatching is beginning in the Boiron (Rubin, pers. obs.).

The degree days (dd, sum of the daily mean temperature values from 1st of March to the date of fish capture) and the number of days with a daily mean temperature ≥15 °C (ndays15) from the 1st of March to the date of the catch were calculated from the logger data.

### 4.4. Statistical Analysis

A total of 697 fish were sampled and used for statistical analysis, which was performed applying the software RStudio (version 2021.09.1, RStudio, Inc, Boston, MA, USA). The correlation between PKD data (percentage of *T. bryosalmonae*-infected fish and infection intensity) and temperature values (dd and ndays15) was assessed with a univariate analysis using Pearson’s coefficient. The significant differences (*p* < 0.05) in the dd and ndays15 means between all *T. bryosalmonae*-infected fish and parasite-free trout were tested with Student’s *t*-test.

## 5. Conclusions

With the anticipated increase in water temperature as a result of ongoing global warming, far-reaching, long-lasting, and, in many cases, dramatic consequences for aquatic ecosystems are to be expected. In particular, PKD of salmonids could become an even greater issue for the survival of many wild brown trout populations. In this light, one of the key challenges facing researchers is to develop a suite of tools for detecting and assessing the impacts of climate change on infectious diseases in complex ecosystems. Therefore, research as presented here is essential to provide a baseline for field studies for a better understanding of the impact of PKD on our wild brown trout populations. For this purpose, long-term temperature parameters, such as the degree days and the number of days with a daily mean temperature ≥15 °C, are a useful tool to compare the influence of temperature across field sites and determine the optimal sampling period for field investigations. This study aimed to follow the temporal variation of PKD manifestation in wild brown trout populations from two rivers with different temperature regimes in order to identify the time window which is most suitable for the robust determination of the disease prevalence. Our findings indicate that water temperature is the main driver for the temporal manifestation of the infection, which, hence, has important implications for the practical design of field investigations. Careful consideration must be given to the choice of the sampling period when aiming to identify the PKD presence in a native brown trout population. Too early, the infection could not have been declared or could have been kept under the histology detection limit. Too late, infected fish could have died due to the infection or parasites could have already been excreted. Average values of ~1500 dd or 30 ndays15 are necessary for having the highest probability to detect histologically *T. bryosalmonae*-infected fish in wild brown trout populations. This threshold should, therefore, be carefully considered when planning sampling campaigns for PKD assessment in a particular stream and could be applied independent of location, level above sea, and weather conditions.

## Figures and Tables

**Figure 1 pathogens-11-00681-f001:**
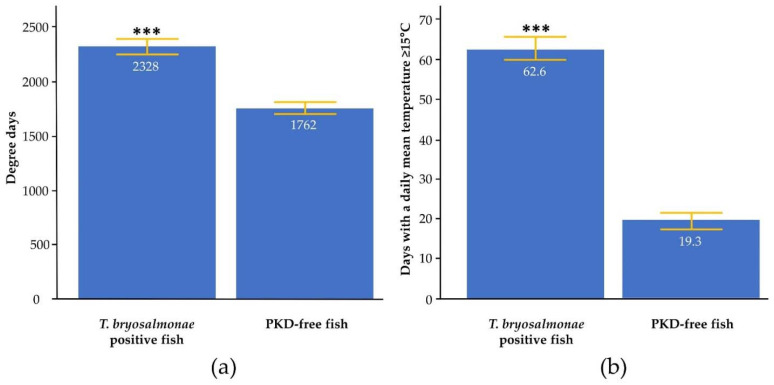
Mean degree days (**a**) and the number of days with a daily mean temperature ≥15 °C (**b**) between *T. bryosalmonae*-infected and PKD-free fish from all sites. White scores indicate means results, yellow lines indicate the standard error, and asterisks indicate levels of significance (*t*-test), *** *p* < 0.001.

**Figure 2 pathogens-11-00681-f002:**
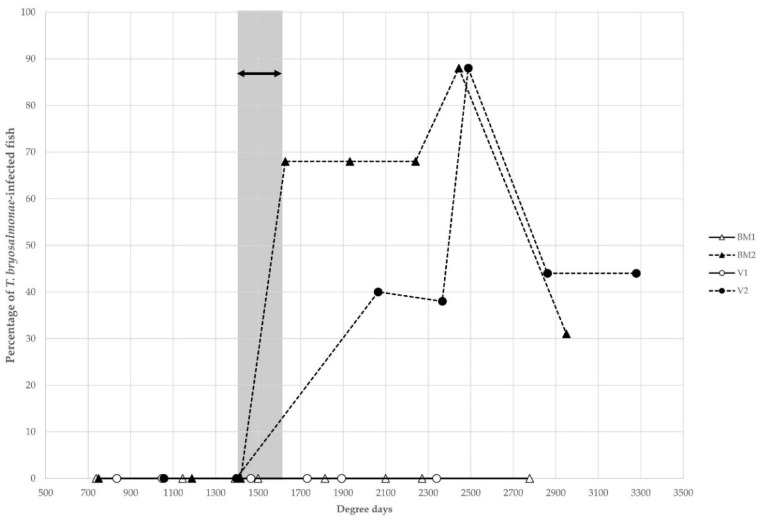
Long-term water temperature measurements expressed as degree days linked with the percentage of *T. bryosalmonae*-infected fish values in the Boiron de Morges (BM) and the Venoge (V) (BM1 and V1 = upstream sites; BM2 and V2 = downstream sites). The grey zone corresponds to the critical degree days threshold between PKD-free and *T. bryosalmonae*-infected fish.

**Figure 3 pathogens-11-00681-f003:**
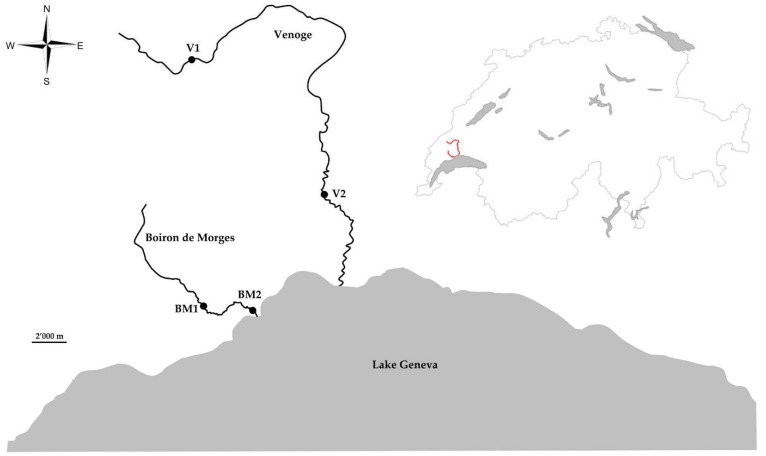
Location of study sites in the Boiron de Morges (BM1 and BM2) and the Venoge (V1 and V2). The two rivers are shown on the map of Switzerland (top right).

**Figure 4 pathogens-11-00681-f004:**
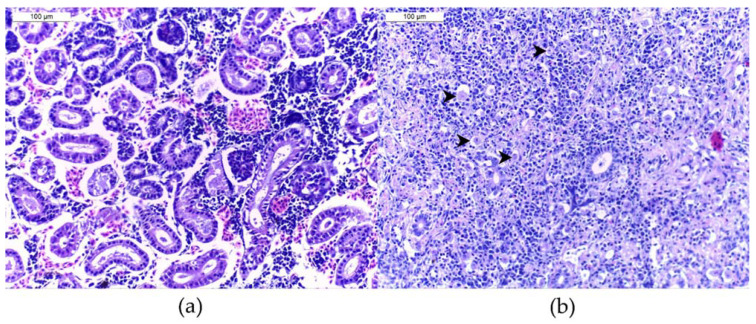
Histological assessment of brown trout *Salmo trutta* posterior kidney H&E-stained slides (**a**) without *T. bryosalmonae* parasite (infection score of 0), (**b**) with a severe infection (score of 6). Four parasites are pointed.

**Table 1 pathogens-11-00681-t001:** Percentage of *T. bryosalmonae*-infected fish based on histological examination and temperature results for the Boiron de Morges (BM1 and BM2) and the Venoge (V1 and V2). Upstream sites = BM1 and V1, downstream sites = BM2 and V2. Grey-shaded boxes correspond to sites in which *T. bryosalmonae*-infected fish were detected based on histological examination.

Station	Year	Sampling Date	Number of Sampled Fish	% of *T. bryosalmonae*-Infected Fish	Infection Severity	Degree Days	Number of Days with a Daily Mean Temperature ≥15 °C
BM1	1	5 June	25	0	0.0	737	0
		8 July	25	0	0.0	1144	1
		22 August	25	0	0.0	1814	18
		12 September	25	0	0.0	2099	20
		8 November	31	0	0.0	2777	20
	2	7 July	2	0	0.0	1391	11
		8 September	25	0	0.0	2316	35
	3	14 July	4	0	0.0	1499	15
		31 August	14	7 *	1.0 *	2271 *	49 *
BM2	1	5 June	25	0	0.0	749	0
		8 July	25	0	0.0	1188	5
		22 August	25	68	3.9	1931	49
		12 September	25	68	3.2	2240	57
		8 November	26	31	1.5	2950	57
	2	7 July	18	0	0.0	1417	28
		8 September	25	88	4.6	2444	88
	3	14 July	25	68	3.3	1627	34
		31 August	25	88	3.5	2488	82
V1	1	19 June	25	0	0.0	834	0
		12 July	25	0	0.0	1047	0
		20 August	25	0	0.0	1465	0
		2 October	25	0	0.0	1892	0
		29 November	26	0	0.0	2339	0
	2	1 September	26	0	0.0	1730	0
V2	1	19 June	25	0	0.0	1056	3
		12 July	26	0	0.0	1397	13
		20 August	25	40	2.9	2064	50
		2 October	25	44	1.8	2661	63
		29 November	25	44	1.3	3278	63
	2	1 September	24	38	2.8	2366	59

* Only one infected fish (infection intensity = 1.0) was found for the first time in BM1. Therefore, it was categorized as a migratory animal coming from an infected downstream zone and was not considered in the discussion.

## Data Availability

Not applicable.

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
