# Peer review of "Reliable Field Assessment of Proliferative Kidney Disease in Wild Brown Trout, Salmo trutta, Populations: When Is the Optimal Sampling Period?"

_pathogens, 2022, doi:10.3390/pathogens11060681_

Round 1
Reviewer 1 Report
I suggest to put a sentence or two in Discussion section about sampling sites e.g. are downstream sites under pollution pressure and information about water quality, if there is any available. Did the authors have chance to check health status of the other fish species during the sampling periods?
Reviewer 2 Report
Please work through the comments provided.

Reviewer 3 Report
This paper describe a 3 year period study concerning the seasonal variation of fish (Salmo trutta) infected by Tetracapsuloides bryosalmonae in two small Swiss streams with difference of temperature regimes between upstream and downstream sites. Rubin and colleagues hypothesized that the number of infected fish, as well as their seasonal distribution, will vary between upper and downstream segment of the two streams. In conclusions, authors state that obtained results are in agreement with this hypothesis. The length and structure of the manuscript are well adapted to the content and shows useful data, however, after deep analysis of the manuscript, this could be accepted after a major revision. These are some comments that authors have to consider for improve the quality of their work.
Keywords please remove wild fish population dynamics
Introduction
Line 42 please change multi-host life cycle in heteroxenus life cycle.
The life cycle of T. bryosalmonae should be better described, please provide more information and bibliographic references. In particular, authors should mention that spore release by bryozoans seems to be undulating, with a peak value in spring/summers.
Line 83 please correct the typing error (ndays15))
Materials and methods
In my opinion, authors should explain how they identified detected parasites. Is well known that T. brysosalmonae provoke pathognomonic lesions, however, for the completeness of the text, is important provide more information about the identification at species level.
Line 244-246 remove this part by M&M, move this sentence to discussions paragraph.
Line 261 please provide the microscope brand and model
Line 262 to 263 authors should provide some pictures of histological sections.
Are macroscopic changes in the inner organs were observed? Did you perform a general anatomopathological examination of sampled fishes?
Line 249-250
Did you record biometric features? Please be more specific.
Have you noticed any differences between the different sizes of sampled fish?
Provide more information and explain if you adopted any criteria for choose fish size during sampling activities.
Line 266-268
Please calculate the prevalence value and cite Bush, A.O.; Lafferty, K.D.; Lotz, M.; Shostak, A.W. Parasitology meets ecology. J. Parasitol. 1997, 83, 575–583.
Discussions
For the readers is important to understand the pathway that conduced authors to formulate the hypothesis proposed in the present study (line 185-188), so, I strongly recommend to better describe how environmental conditions affect the life cycle and transmission of T. bryosalmonae to fish host.
Reviewer 4 Report
Salmo trutta populations decline is a global problem of freshwater ecosystems, particularly in rivers and streams of mountainous topographies. Under this prism the topic of the study is interesting. Nevertheless, the main drawback is the absence of molecular confirmation of PKD detection. The authors estimated a mean of degree days (days with temperature over 15 ºC) for the histological detection. It would be however more informative to examine how many molecularly positive samples were not detected histologically, in order to calculate a more realistic detection threshold for histological detection.
Also, some important comments that have to be addressed are presented in the following lines:
Line 20: “of the percent of”, maybe replace with “the prevalence of” or the “infection rate”?
In lines 32-34, I would expect to see the main recommendations regarding the sampling campaign, at least the most important one that was concluded.
A main reason for the decrease of Salmo trutta populations is the genetic admixture of different populations due to illegal translocations (Giantsis et al. 2022, https://www.mdpi.com/2076-3417/11/19/9034/htm), a phenomenon that contributes also to the dispersal of parasites. This information should be added in the Introduction as well.
Since the authors claim in the introduction that the two investigated streams host PKD-positive brown trout populations, a relative reference should be included. Otherwise the scope of the study has to be modified towards the investigation of the presence of the parasite in these population. I suggest to better add a reference for the previous detection of this parasite in these two streams.
Is there any figure from the histological results of PKD detection? Since only this was the methodology, the authors should add a photo from histological procedures
There is not even one reference in the first paragraph of the discussion, although the authors present previously reported inferences. References should be added in lines 180-182, 182-184, 184-185 etc.
Round 2
Reviewer 2 Report
Responses to editorial questions are satisfactory.

Author Response
Reviewer comment: Responses to editorial questions are satisfactory.
Reviewer 3 Report
After an in-depth analysis of the manuscript, I declare that the authors have made all the necessary changes, only need a few minor corrections. Please see below for more detailed comments.
Line 42 please correct the typing error
Line 52 please report some susceptible salmonid species
Line 286-288. Was the class size of the fish sampled selected based on a previous study conducted in Switzerland? Please provide a bibliographic reference or specify that this choice come from your unpublished data.
Reviewer 4 Report
The authors addressed all the comments
Author Response
Response of the Reviewer: The authors addressed all the comments.